# Hypertemporal Imaging Capability of UAS Improves Photogrammetric Tree Canopy Models

**Andrew Fletcher [1],* and Richard Mather [2]**

1   Science and Engineering Faculty, Queensland University of Technology, Brisbane 4000, Australia
2   School of Business, Law and Computing, Buckinghamshire New University, High Wycombe HP11 2JZ, UK; richard.mather@bucks.ac.uk
*   Correspondence: a20.fletcher@qut.edu.au; Tel.: +61-408-249-953

**Abstract:** Small uncrewed aerial systems (UASs) generate imagery that can provide detailed information regarding condition and change if the products are reproducible through time. Densified point clouds form the basic information for digital surface models and orthorectified mosaics, so variable dense point reconstruction will introduce uncertainty. Eucalyptus trees typically have sparse and discontinuous canopies with pendulous leaves that present a difficult target for photogrammetry software. We examine how spectral band, season, solar azimuth, elevation, and some processing settings impact completeness and reproducibility of dense point clouds for shrub swamp and Eucalyptus forest canopy. At the study site near solar noon, selecting near infrared camera increased projected tree canopy fourfold, and dense point features more than 2 m above ground were increased sixfold compared to red spectral bands. Near infrared (NIR) imagery improved projected and total dense features two- and threefold, respectively, compared to default green band imagery. The lowest solar elevation captured (25°) consistently improved canopy feature reconstruction in all spectral bands. Although low solar elevations are typically avoided for radiometric reasons, we demonstrate that these conditions improve the detection and reconstruction of complex tree canopy features in natural Eucalyptus forests. Combining imagery sets captured at different solar elevations improved the reproducibility of dense point clouds between seasons. Total dense point cloud features reconstructed were increased by almost 10 million points (20%) when imagery used was NIR combining solar noon and low solar elevation imagery. It is possible to use agricultural multispectral camera rigs to reconstruct Eucalyptus tree canopy and shrub swamp by combining imagery and selecting appropriate spectral bands for processing.

**Keywords:** UAS; canopy; repeatability; photogrammetry; eucalyptus; shrubs

## 1. Introduction

Modern photogrammetric processing software, high-precision global navigation satellite systems (GNSSs), high-resolution consumer cameras, and mobile computing power have combined to place remote sensing imagery and photogrammetry products in the hands of field ecologists and environmental scientists [1,2]. Tree and vegetation monitoring are commonly proposed and highly cited applications of small uncrewed aerial systems (UASs) [1–5]. Software automation of complex processing workflows has provided accessible, low-cost photogrammetric outputs to researchers and industry (Figure 1). Routine monitoring requires that measurements are repeatable and reproducible through time for features of concern [6]. The reproducibility of equivalent products from independent imagery or software using automated photogrammetry processes is currently uncertain [7–9]. In this study, tree canopy extent is a core monitoring requirement as the target communities are typically treeless [10] and currently indirectly mapped by the boundary of the surrounding forest canopy [10,11].

The broad processing steps to generate photogrammetric outputs from UAS imagery sets is similar for all processing software platforms (Figure 1). The operator determines the choice of sensor, flight path, overlap, environmental conditions, and study target properties (Figure 1a). These decisions include proportional overlap [12–14], camera resolution [14,15], camera attitude [15,16], and illumination [7,12,17,18], which all affect both point cloud and raster outputs. The initial processing of imagery involves many programmatic decisions and processing steps (Figure 1b). Briefly, these include image feature description and matching in overlapping images, calculation of aerial triangulation and camera geometry, incorporation of ground reference or other geolocation data, and bundle block adjustment to optimise the camera distortion model and location uncertainty (Figure 1b). Most of the algorithms and settings in Figure 1b are proprietary in all commercial software packages. Once camera location and pose is established pixel matching photogrammetric algorithms are applied to stereo or multiview image blocks to reconstruct a dense point cloud (Figure 1c). The outputs of Figure 1c are fundamental to many aspects of research in forestry. Dense point clouds are often compared with UAS [19] or airborne laser scanning (ALS) [20,21] products and potentially provide low-cost forest inventory [22–24]. Raster-based outputs are generated by filtering and interpolating the dense point cloud to create a digital surface model (DSM) (Figure 1d). Digital surface models (Figure 1d) are often proposed to characterise tree canopy properties [25–29] or correct bidirectional reflectance distribution [30]. Individual images are orthorectified by combining camera location and pose with a digital surface model. Finally, orthomosaic rasters are created by selecting and/or blending spectral information from available images covering a given area (Figure 1e). Orthomosaic rasters can provide spectral information to classify tree species [31,32]. The completeness and reproducibility of dense point clouds therefore directly impact any multitemporal analysis of dense point clouds, digital surface models, and orthomosaic raster outputs.

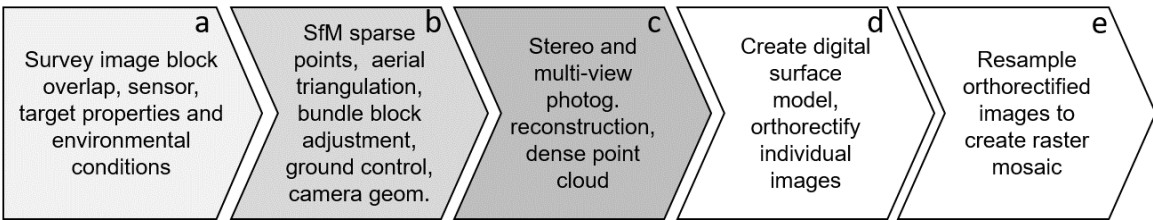

**Figure 1.** The major steps required to create an orthrectified image mosaic include (**a**) operator decisions and sensor; (**b**) structure from motion (SfM) and iterative algorithms to reconstruct camera pose; (**c**) photogrammetric matching algorithms; (**d**) filtering and surface model reconstruction; (**e**) raster prioritisation, resampling and/or blending.

The choice of platform constrains the image collection method, sensor, and target geometry [33]. Software [7,9,34,35], software settings [36], camera station forward overlap [12,14], spatial resolution [14,15], camera attitude [15,16], and illumination [7,12] are just some of the factors that interact to determine the outcomes of image-processing workflows. Point cloud filtering algorithms and digital surface reconstruction methods will also influence orthomosaic raster products [30]. UAS imagery processing publications commonly use independent measurement techniques such as airborne LIDAR [37–42], UAV LIDAR [19,43], GNSS field observations [38,44,45], and field biophysical measurements [9,21,25,28] to assess the accuracy of products. However, the impacts of target properties and environmental and sensor conditions are complex and poorly understood [14], and many multi- and hyperspectral sensors rely on preflight calibration [9] or post-processing normalisation [31,46]. Empirical determination of appropriate imagery collection parameters is thus extremely complex given the multiple potential sources of variability that exist in automated photogrammetry image processing pathways [12–14]. Tree canopies are complex objects to model and translate to a 2D raster. Eucalyptus canopies may be discontinuous or sparse [8], shaded or variably illuminated, and move in even light winds [47]. Movement prevents logical feature matching between images, while visual

porosity and feature shape and texture change with perspective shift limits matching across camera field of view. Variations in the dense point cloud models potentially alter the subsequent surface models and orthomosaic raster outputs. However, these models are rarely interrogated for major sources of error, and for reasons of convenience, default options are often utilised in developing products for vegetation modelling from photogrammetry [9,30].

While numerous processing decisions are undertaken in reconstructing a dense point cloud, default processing options are often cited for vegetation modelling from photogrammetry [9,30]. While it is acknowledged that spectral properties of vegetation will affect resulting feature registration [9], shrub and crop vegetation targets are either reconstructed using default imagery processing settings [18] or rely on the high spatial resolution of visible colour cameras to reconstruct 3D structure for lower-resolution spectral signatures [31,48]. Softwood forestry has seen substantial progress in the application of consumer colour and hyperspectral imaging to develop traditional forestry metrics such as growth rates [49,50] and above ground biomass, typically based on canopy height models [20,23,51]. Tree canopy sparse point cloud density is positively correlated with forward overlap for broadleaved tree canopies, independent of spatial resolution [12], although percentage overlap in this instance was covariant with flight altitude due to interval camera triggering while flight speed was constant. Decimating high-interval imagery demonstrates that both canopy model and ground-based feature registration in dense models are improved with increasing forward overlap [14]. Feature registration relies on features being identified and matched in multiple images. Low overlap percentages therefore exclude features by occlusion through a wide perspective angle, reduced sampling, and reduced feature similarity due to perspective range that combine to decrease visibility and the probability of registration. Repeatability and robustness of dense point clouds has been demonstrated by decimation of an oversampled imagery set [13,14]. This approach ensures that environmental variables due to time of collection are normalised across the compared products. Addressing illumination differences or camera station variability requires the comparison of independently captured imagery sets containing differences in illumination and camera station placement across a common unchanged target.

Eucalyptus typically have discontinuous canopies with pendulous leaf orientation that move individually and as independent canopy clumps with wind [47]. Near infrared (NIR) spectral reflectance exceeding 90% has evolved to reduce solar energy load [52]. Larger Eucalyptus trees in savannah ecosystems achieved 70% identification compared to airborne LIDAR when imagery was collected by a consumer camera with NIR filter removed [42]. Young and smaller trees typically have very sparse canopies [8,47]. Smaller Eucalyptus canopies are often poorly modelled [42] or absent [8]; however, it has been concluded that NIR imagery enhances tree detection [8,42]. Discontinuous Eucalyptus forest canopies in temperate regions are reconstructed in close range colour image photogrammetry (<15 m) [19]. In this instance, UAS LIDAR provided independent measurement showing photogrammetric occlusion of upper canopy branches and sub-canopy ground features [19]. Other studies of canopy structure in Eucalyptus examine forestry plantations where tree canopy is light-limited and conical in shape and UAV photogrammetry products are compared to independent methods at a single time [21,53,54]. Appropriate minimum specifications for sensor, environment, and processing to achieve reproducible Eucalyptus canopy models are currently uncertain. It is possible to compare feature registration in equivalent imagery sets by processing individual spectral bands from multispectral multicamera rigs (e.g., Parrot Sequoia, MicaSense Red Edge). The analyst is able to select a master camera from this rig for photogrammetric purposes and subsequent filtering of the dense point clouds to refine a DSM for orthomosaic imagery sampling. Illumination and environmental variability may be examined by employing UAS ability to capture a target location several times within a single day or 24 h period.

Monitoring is primarily concerned with change detection and requires limited measurement uncertainty and transferable methods [6]. The study sites are located in shallow sloping drainage lines, providing a range of natural topographic relief and vegetation types. Imagery was collected at a consistent camera station density and altitude range five different times in consecutive winter and

summer seasons. We examine dissimilarities between dense point clouds, and particularly those of tree canopy features as they may disproportionally affect final products by introducing steep-to-vertical faces in surface models or fail to register canopy structure completely. Our imagery sets provided variation in environmental (sun elevation and azimuth, target sensor geometry, season) and spectral (red, green, red edge, near infrared) properties for dense point cloud reconstruction of a fixed target area containing equal proportions of shrub land and Eucalyptus forest. Programmatic variability in dense point clouds was minimised by applying consistent processing settings in a single software package. Multispectral images were processed as individual spectral bands and multicamera rigs. Different spectral bands were set as the master in a multicamera rig to compare outputs of dense point clouds. Imagery processing was limited to sparse feature registration of camera station and point cloud densification. All subsequent analyses were conducted directly on the point clouds to limit the impacts of photogrammetry software filtering and interpolation algorithms on final products.

*Main Aims*

This study investigates several of the factors that affect the reconstruction of tree canopy structure in natural forests to identify methods for developing reproducible tree canopy models. This paper addresses two primary questions:

1. Does choice of spectral band in multispectral multicamera rigs result in variable tree canopy feature reconstruction?
2. Are photogrammetric reconstructions of dense point cloud features altered by solar illumination conditions in a natural Eucalyptus forest and shrubland landscape?

## 2. Material and Methods

### 2.1. Site and Target Properties

The study site is located within the Newnes State Forest, approximately 100 km Northwest of Sydney and 12 km North-Northeast of Lithgow (Figure 2). The target vegetation community is characterised by sclerophyllous shrub, sedge, and grass tussocks in shallow sloped (< 4%) drainage lines on sandstone [10]. The communities are subject to a range of impacts including forestry, high-voltage power line clearing, recreational off-road vehicles, and underground mining subsidence. Imagery was collected in winter (24 August 2017) and summer (17 January 2018). The surrounding vegetation is eucalypt forest. Monitoring and rehabilitation are required activities for commercial users who may impact these communities. Encroachment of Eucalyptus trees has been identified as an indicator of negative impact.

Existing Eucalyptus tree extents were determined around the mapped target community (approx. 5.58 ha) using a 20 m buffer that resulted in a total study area of 9.87 ha (Figure 2). The target is predominantly vegetation with bare ground and shrub layers in the understory of the forest and some small patches of bare ground in the target community (Figure 3). Biophysical properties of forest canopies such as canopy density, branch structure, and leaf orientation will alter the texture and pattern of imagery and may drive variable reconstruction of dense point clouds. Eucalyptus trees in natural forest environments are poorly studied but, conceptually, represent a range of biophysical factors that are likely to increase the complexity of sparse and dense point reconstructions.

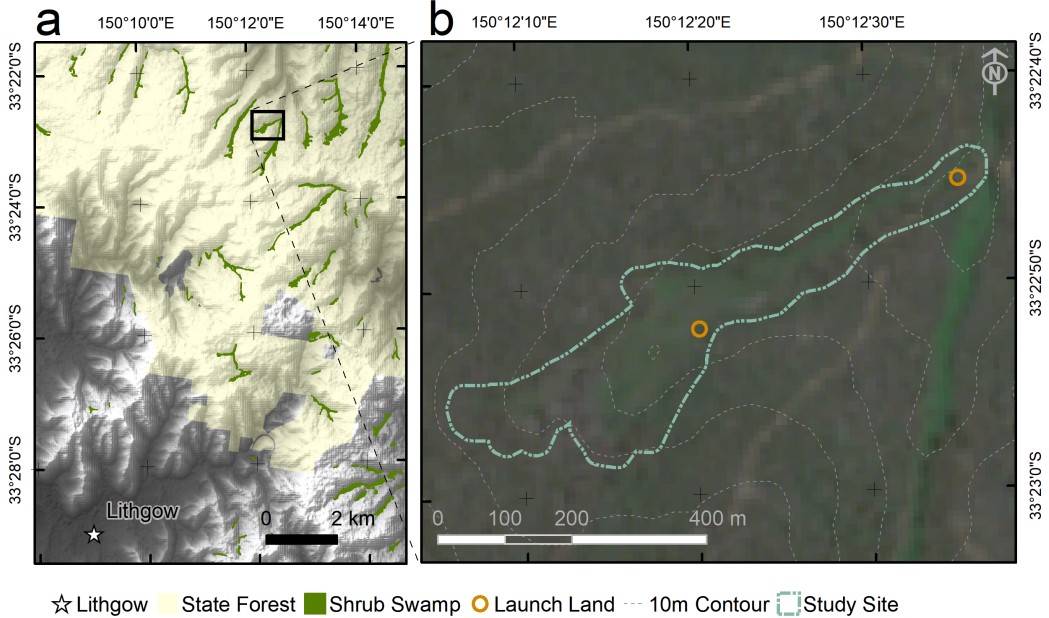

**Figure 2.** (**a**) Location of the study site relative to Lithgow, Newnes State Forest, and other mapped shrub swamp communities shown with a hillshade digital elevation model (USGS Shuttle Radar Topography Mission v2). (**b**) Study site extent showing launch and landing areas and 10 m contours. Background true colour Sentinel 2 imagery acquired 14 February 2018 (Credit: European Union Sentinel Data).

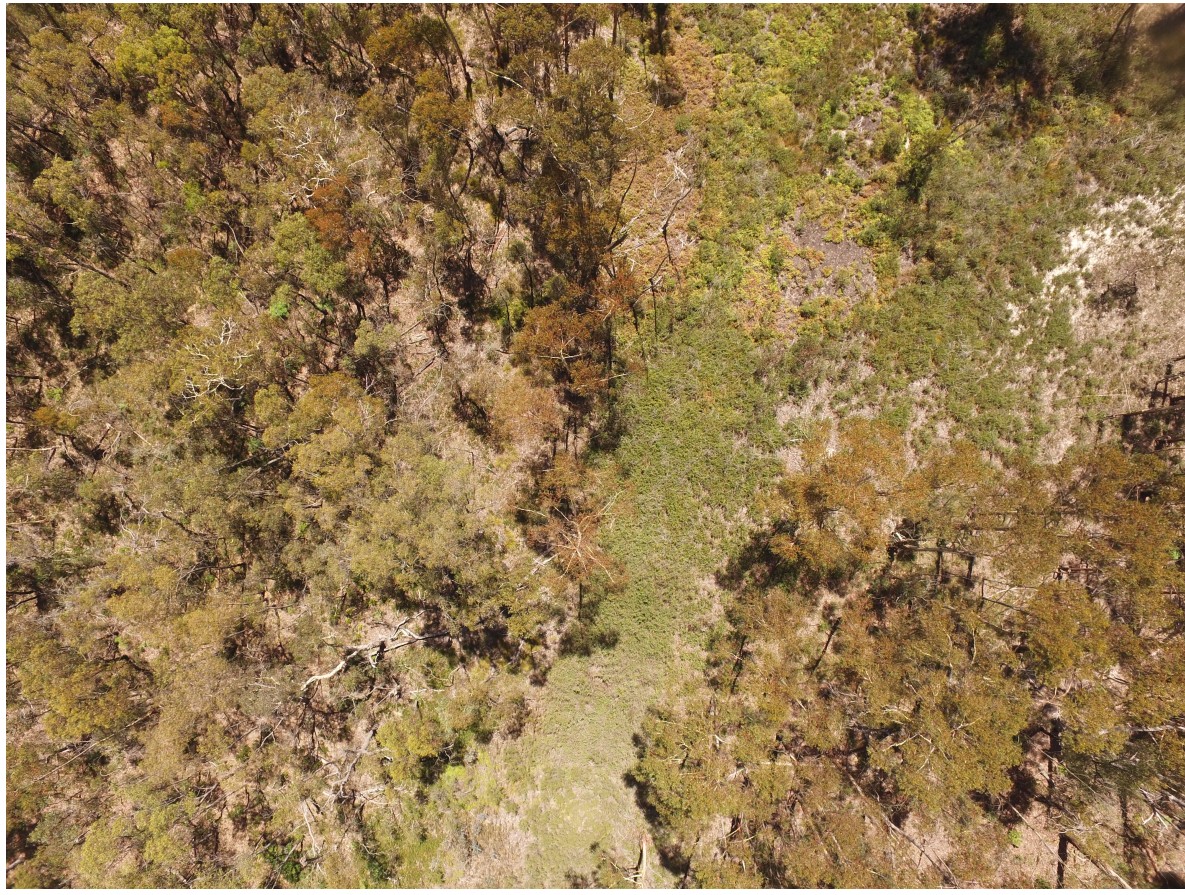

**Figure 3.** Image of the target community and fringing Eucalyptus forest acquired during a summer midday flight. Average resolution approximately 3 cm at ground level.

Environmental Conditions

Both time of day (see Table 1) and the orientation of the long axis of the community ensured a wide range of illumination conditions were captured during safe operating hours. The low landscape position of the launch sites compared to surrounding forest canopy prevented local records of wind speed and direction to be recorded for the study site, so the average conditions at the nearest official meteorology station are reported (Table 1).

**Table 1.** Dates, times (UTC), sun azimuth, and elevation for first and last images collected in each survey. Start (S) and end (E) illumination conditions were calculated for a flat location at the latitude and longitude of the study site. Wind conditions were recorded in Lithgow, New South Wales, approximately 12 km away and at 300 m lower elevation.

| Season | Time | Date (UTC) | Wind (9am m.s$^{-1}$) | Time (UTC) | Bracket | Elevation (°) | Azimuth (°) |
|---|---|---|---|---|---|---|---|
| summer | T1 | | | 2100 | S | 21.77 | 101.33 |
| | | 20180116 | 3.1 | 2135 | E | 28.8 | 97.2 |
| | T2 | | | 2315 | S | 61.8 | 70.95 |
| | | | | 0006 | E | 71.2 | 52.2 |
| | T3 | 20180115 | 1.1 | 0425 | S | 45.1 | 272.8 |
| | | | | 0503 | E | 37.2 | 267.5 |
| winter | T1 | | | 1854 | S | 27 | 53.29 |
| | | 20170824 | 1.9 | 1958 | E | 36.5 | 38.9 |
| | T2 | | | 2252 | S | 43.8 | 342.5 |
| | | | | 2346 | E | 39 | 326.3 |

## 2.2. UAS, Sensor, and Survey

All imagery was collected by a contracted commercial UAV operator within legal guidelines for UAS operations in Australian airspace. A modified DJI Inspire 1 (Shenzhen, China) carried both the 12 Mp colour camera (DJI X3) and Parrot Sequoia simultaneously. Flight planning for DJI autopilots uses the launch location to determine target altitudes. All flights were planned for and controlled by the Inspire 1 autopilot. All flights were conducted at a target of 80 m above launch points. Two locations were selected to minimise topographic relief while maintaining visual line of sight throughout flights (Figure 2). Camera station elevation varied between 70–90 m above ground level (AGL). Geolocation data for multispectral imagery was collected by the Parrot Sequoia. The Parrot Sequoia is a low-cost, four-band, five-camera rig that includes a 16 Mp RGB camera and four narrow-pass single-band 1.2 Mp sensors (green: 530–570 nm, red: 620–680 nm, red edge: 730–740 nm, (near infrared) 770–810 nm). The multispectral sensor was mounted in a fixed downward landscape orientation on the central airframe of the Inspire, while the downwelling sensor was mounted directly above the camera. All survey flight lines were separated by approximately 19 m (80% side lap). The density of camera stations therefore vary according to platform forward velocity. The 1 Hz interval is the fastest sustainable rate of capture for the Sequoia and produced an endlap of 87% or 8–9 m in line intervals. Platform forward speed was set by the need to achieve flight line coverage of the target area within three 10 min flights. The four individual sensors in the rig store images as individual band files, resulting in four images at each camera station. Because the camera is independently controlled at a 1 Hz interval, imagery is collected throughout the flight. This means that non-nadir imagery was captured when the platform turned at the end of survey legs. Images collected during take-off and landing were manually removed from each image set, and only images collected at target altitude when moving were retained. Sequoia imagery was captured as 11 bit TIF images with corresponding downwelling spectral irradiance, GPS coordinates, and attitude metadata provided by the sensor. Example images captured near the coordinates of Figure 3 are presented for summer morning and midday (Figure 4). The impact of tree canopy spectral reflectance profile is shown with a red edge and NIR imagery having high brightness values. Tree canopy features appear muted in the red spectral bands, and bare ground features show clearly in imagery captured near solar zenith (Figure 4).

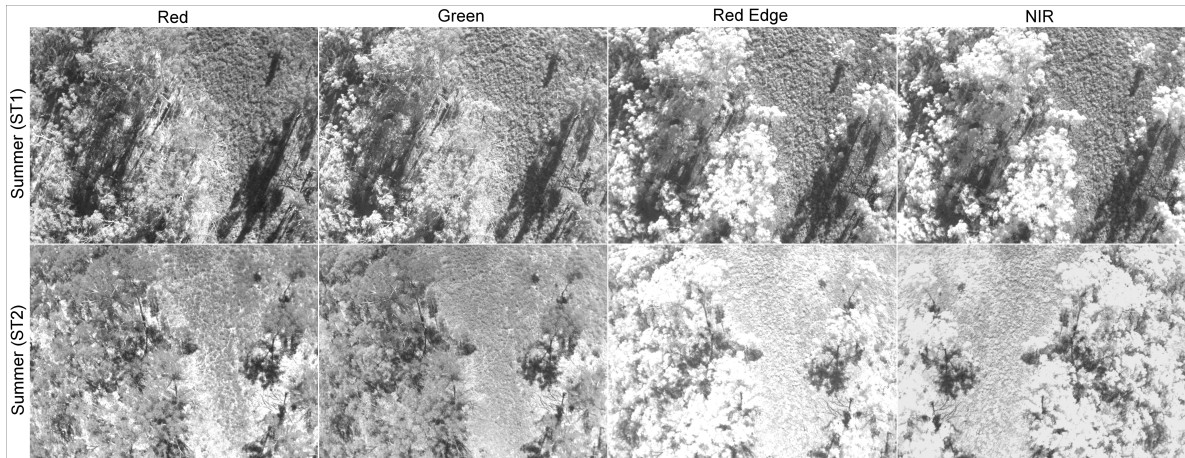

**Figure 4.** Single spectral band images (columns left to right) for red, green, red edge, and near infrared (NIR) cameras captured in summer during morning (top row) and midday (bottom row) flights.

The calibration of camera station location and internal and external camera geometries required feature matching between overlapping images. Perpendicular survey grids are not feasible given the corridor geometry of the target site (Figure 2). Flight plans were therefore survey lines oriented to the long axis of each section of the community. A total of three 10–12 min flights within a 2 h window provided complete coverage of the community at two times of day (T1 & T2) during winter and three times of day during summer (T1–T3). Additional time between flights was required to allow movement between suitable locations to maintain visual line of sight with the platform and to capture imagery using other sensors. The total number of camera stations varied by 96 (12%) depending on the specific flight path planned by the operator (Table 2). The autopilot of the UAS used for this project sets flight altitude relative to launch site elevation, variation in launch location, and GPS accuracy at the time of flight planning, resulted in different absolute flight elevations. Final camera elevations are reported from calibrated camera positions after initial processing including survey-grade ground control (Table 2). Different camera station geometry between image projects may affect final products. The average 3D distance between all camera stations in all imagery sets was calculated (Table 2). Within season, 3D camera station locations varied between 3.2–4.8 m but was higher between seasons 7.5–9 m. Elevation of sensor varied by approximately 7 m, or 9% (Table 2).

**Table 2.** Number of camera stations processed per spectral band, range of camera station elevations (90% of all cameras) for each project in Australian Height Datum (AHD) and average 3D distance to the nearest camera station ($\pm$1 SD) for all monitoring combinations.

|  | Time of Day | Stations | z (m) 90% | Summer | | | Winter | |
|---|---|---|---|---|---|---|---|---|
|  |  |  |  | T1 | T2 | T3 | T1 | T2 |
| Summer | T1 | 740 | 1141.6–1144.6 |  | 3.4 $\pm$ 2.1 | 4 $\pm$ 2.5 | 8.6 $\pm$ 3.3 | 8.9 $\pm$ 4.9 |
|  | T2 | 732 | 1142.7–1146.3 | 3.2 $\pm$ 1.3 |  | 3.9 $\pm$ 2.1 | 8.6 $\pm$ 3.7 | 8.8 $\pm$ 4.9 |
|  | T3 | 787 | 1143.1–1146.7 | 4.7 $\pm$ 4.4 | 4.8 $\pm$ 4.5 |  | 8.9 $\pm$ 4.5 | 9.0 $\pm$ 4.9 |
| Winter | T1 | 733 | 1135.5–1149.5 | 7.9 $\pm$ 2.2 | 7.8 $\pm$ 2.6 | 7.6 $\pm$ 3 |  | 4.4 $\pm$ 3.6 |
|  | T2 | 691 | 1135.3–1149.3 | 7.7 $\pm$ 2.7 | 7.6 $\pm$ 3.2 | 7.5 $\pm$ 3.3 | 3.9 $\pm$ 2.3 |  |

*2.3. Ground Control*

Six semipermanent ground control marks (GCM) were installed in August 2017, and locations were determined using GNSS logging control marks (Propeller Aeropoints) that were post-processed against base station logs. Greatest variance in absolute position was less than 27 mm in X, Y using EPSG 28356 and Australian Height Datum (Z) coordinate systems. The same markers were used for both seasons after confirming that the location was unchanged. High-accuracy ground control is essential when comparing products collected with low-precision GNSS receivers such as UAS

autopilots and small mobile devices. The distance to the nearest GCMwas calculated for the target area using Euclidean planar distance, as the overall slope for the site was 3.8% (Figure 5b). Half of the total buffered area was within 72.5 m of a GCP and 95% was within 91 m (Figure 5a). Camera footprints at ground level were approximately 96 m perpendicular to and 74 m along flight lines.

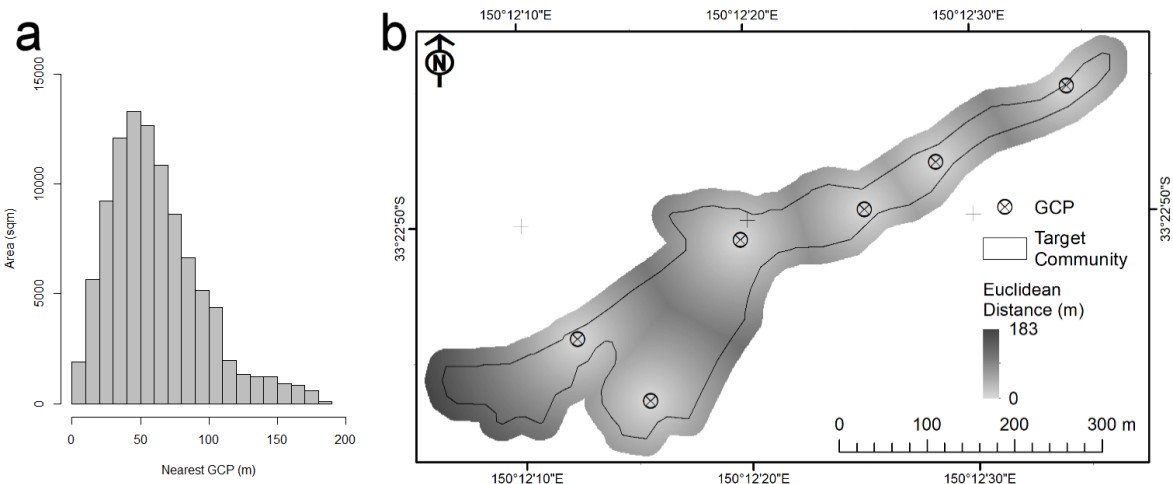

**Figure 5.** (**a**) Histogram of distance to nearest GCM mark (10 m bins) for 1 m$^2$ cells covering the buffered study site. (**b**) Locations of each ground control point with distance.

### 2.4. Imagery Processing

Image sets were processed using Pix4Dmapper versions (4.2.6–4.3.33). The image processing settings in Table 3 refer to steps 2 and 3 in Figure 1 because the dense point cloud output from step 3 (Figure 1) determines both DSM and orthomosaic composition. Multispectral sensor image sets were processed as both multicamera rig and as individual spectral bands to determine impact on the extent and consistency of detected canopies.

**Table 3.** Processing parameters used in Pix4D. Parameters not presented were not applied.

| Initial Processing | |
|---|---|
| Keypoints Image Scale | Full |
| Matching Image Pairs | Aerial Grid or corridor |
| Targeted Number of Keypoints | Automatic |
| Calibration Method | Standard/All internal external parameters |
| Rematch | Automatic |
| **Point Cloud Densification** | |
| Image Scale | 1 (Original image size, Slow) & multiscale checked |
| Point Density | High (slow) |
| Minimum Number of Matches | 3 |
| Matching Window Size | 7 × 7 pixels |
| Image Groups | Green or NIR |
| Point Cloud Filters | Use processing area, use annotations checked |

#### 2.4.1. Initial Processing

All flights were conducted in survey lines with the sensor near nadir, so matching was set to an aerial grid. Perpendicular grids were not attempted due to flight duration and the corridor nature of the community shape. Keypoint generation identifies robust computer-described features within images that may be matched between overlapping images with full scale representing at pixel resolution. Combining these features with geolocation information from the sensor and ground control allows camera geometry and pose to be described. This setting was chosen because the Parrot Sequoia has

a 1.2 Mp sensor and the target is almost completely covered by vegetation that poorly adheres to photogrammetric feature requirements. Global average image keypoint count for initial processing across the thirty projects was 31,218 ± 5244 (1 SD). The keypoint match number was set to automatic as high values are unlikely to be achieved in vegetation targets with 1.2 Mp sensors. This resulted in an average of 7711 ± 1963 (1 SD) matched keypoints per image across thirty projects. Setting camera internal and external parameters may reduce processing times and improve accuracy but assumes that the camera is not affected by vibration and shocks associated with field operations. Full calibration in each project was therefore selected. Rematching of features ensures consistency of initial processing and reduces noise associated with mismatched features. All projects recorded 99% or 100% of cameras calibrated at the initial image-processing stage (Figure 1b).

2.4.2. Point Cloud Densification

Feature matching was performed at pixel resolution as the sensor is limited to 1.2 Mp. Point density was set to high as this study was primarily concerned with dense point cloud reconstruction. Alternative settings provided were low/fast and optimal. The optimised goal was not stated but is likely associated with construction of orthomosaic products. In photogrammetric dense point cloud construction, gaps between matched features are interpolated. The interpolation distance is controlled by the matching window size, so this was set to the minimum possible at $7 \times 7$ pixels or centre pixel with a three-pixel buffer. This setting is important as it determines the distance at which two features should be to be considered unique in independently constructed products. Reconstructed features more than 6 pixel GSD apart were considered to be separate features.

*2.5. Dense Point Cloud Preprocessing and Analyses*

Automated photogrammetric processing relies on feature description and matching algorithms. Differences in feature spectral reflectance will cause different features to be matched depending on the spectral band of the images processed. Multi-camera multispectral camera rigs with separate sensor planes and lenses require a master camera to be selected; dense point cloud reconstruction is developed with this camera to ensure a consistent digital surface model is created. Orthorectification is then conducted for images from other cameras using the measured perspective shift of the camera location and the location and pose of the master camera. Dense point clouds were constructed using individual cameras of the multispectral camera and the multicamera software default setting (green band master) and multicamera with NIR as master. Dense point clouds were processed in the coordinate system used for ground control (EPSG:28356). All dense point cloud products presented are viewed from the top without perspective to allow visualisation of dense feature elevation relative to point cloud structure. Point cloud processing, statistical outlier removal, and rasterisation were conducted using CloudCompare v2.10.

The average pixel spatial resolution for the thirty projects was 7.96 ± 0.25 cm (2.96 standard deviations). The feature matching window size was set to the minimum of $7 \times 7$ pixels, providing a potential error of 24 cm from a verified dense point. Feature reconstruction required at least three images to reduce interpolation effects. Dense points from independent projects that are separated by more than 48 cm in 3D space can be considered to represent distinct reconstructed features at 6 pixels of separation. Following a cloud-to-cloud 3D Euclidean distance calculation, dense points more than 0.5 m from an independently generated dense point cloud were extracted. Small groups of outlying dense points were removed using 6 pixels at 1 SD (Statistical Outlier Removal tool in CloudCompare). These small groups or single points predominantly result from interpolation and small 3D offsets between independently processed dense clouds. The remaining dense points were considered features that were not reconstructed in both projects and were therefore inconsistently reconstructed. Dense point rasterisation was applied at a minimum mapping unit resolution (0.16 m) to visualise the presence of dense points and minimum and maximum dense point elevation within each cell. This process allowed different spectral bands within a single flight to be compared for the

reconstruction of both ground and tree canopy features. To compare spectral and temporal differences in feature reconstruction, dense points of one cloud separated by more than 50 cm from the comparison cloud were displayed with the combined points of both clouds where separation was less than 50 cm. A simple vertical perspective was used to identify upper surface features that are reconstructed in each product. The structural component of tree canopy in the study site ranges over approximately 40 m for the tallest trees. Features reconstructed may include ground cover vegetation, ground, trunks, branches, or leaves. The shrub- and tussock-dominated vegetation in the target community (Figure 3) was removed by sampling the dense point cloud at a spatial interval that extends beyond individual shrubs. The digital elevation model was calculated to a 2 m spatially decimated minimum elevation using the red spectral band (see Results Section 3.1.1) as a ground layer. Products processed from the red camera provided increased completeness of ground level (bare earth/dead vegetation features) and reduced the number of reconstructed tree features. The resulting minimum z points were used to define the ground level for the purposes of this study. Cloud-to-cloud Euclidean distance was calculated with x, y, and z components separated to calculate the point elevation difference to the red digital elevation model. Point z differences were then plotted as histograms to describe the frequency and numbers of features that were inconsistently reconstructed.

## 3. Results

### 3.1. Point Cloud Models

Total reconstructed dense point clouds for thirty independently processed projects were compared across the fixed processing area. Total dense reconstructed point number varied by more than 10 M points (Figure 6). Red edge and NIR imagery and low solar elevations reconstructed almost 20 M dense points more than 2 m AGL. Red and green band imagery captured near solar noon reconstructed fewer than less than 5 M. Solar elevation was negatively correlated with > 2 m AGL reconstructed points for both green, red, and green band master products (Figure 6). The proportion of total reconstructed dense points representing trees (> 2 m AGL) was lowest in all spectral bands for the ST2 imagery set (5.8%–29.4%), which coincided with maximum solar elevation (Table 1). Red, green, and green camera master products reconstructed fewer tree canopy dense points compared to red-edge and NIR for all solar conditions (Figure 6). Low solar elevation coincided with a greater total number of dense points for all cameras and multicamera projects. Tree canopy dense point clouds exceeded 25% of the total cloud number for red edge and NIR image sets, regardless of solar elevation and for all spectral bands at the lowest elevation (ST1).

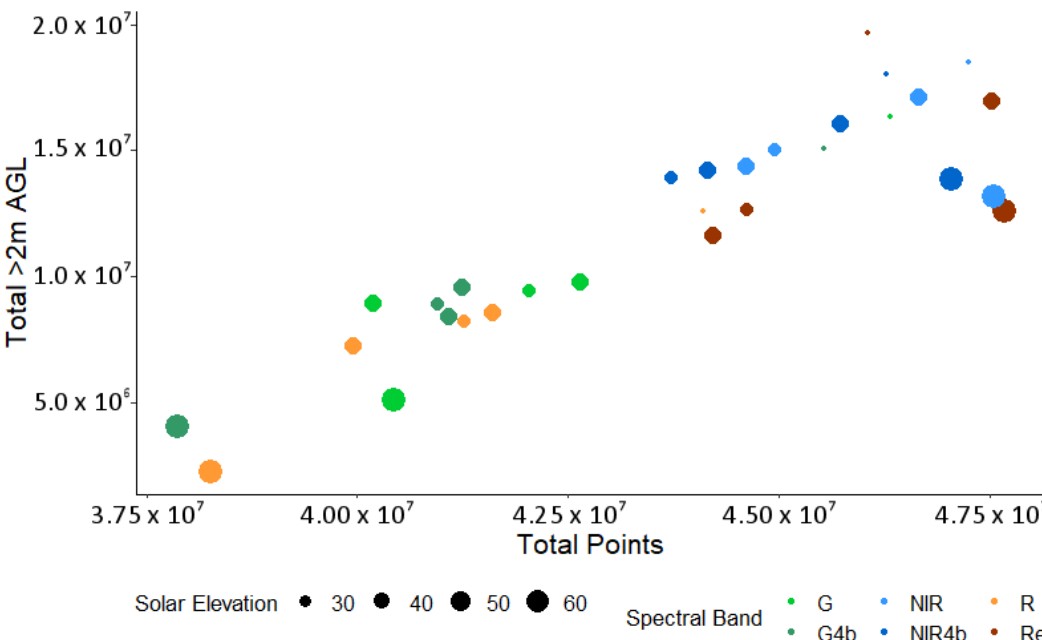

**Figure 6.** Total dense points in final project compared to total points more than 2 m above ground level (AGL). Processing was conducted for individual cameras (G, green; R, red; NIR, near infrared; red edge, Re) and as a multicamera rig with green (G4b) and NIR (NIR4b). Symbol size is directly related to solar elevation. See specific ranges for each flight in Table 1.

### 3.1.1. Image Spectral Properties

Solar zenith is typically targeted for remote sensing purposes to improve radiometric properties and reduce shadows that disproportionally affect high-resolution remote sensing. Because the Parrot Sequoia captures simultaneous images in four spectral bands, the impact of spectral band was examined by processing each camera image set separately using identical settings. Minimum and maximum point elevations were recorded for a 0.16 m pixel resolution across the study area to determine completeness and nature of the upper and lower limits of dense clouds. By subtracting minimum and maximum elevation rasters for each camera, it is possible to identify tree feature reconstruction as a vertical dense point distribution greater than 2 m. At the highest solar elevation range (summer T2), red camera dense point clouds reconstructed 1664 m$^2$ of projected dense canopy point features (Figure 7a,b). The projected area of tree canopy dense registration increased to 2942 m$^2$ in the green spectral band (Figure 7c,d), while red edge (6292 m$^2$) and NIR (6854 m$^2$) dense point clouds reconstructed the greatest area of tree canopy in the study area (Figure 7f). NIR spectral band reonstructed a four fold greater projected area of tree canopy compared to the identical set of red camera imagery. Total coverage of the target area or completeness of representation for the study area was determined as dense point projected coverage at the minimum mapping unit (0.16 m). Gaps within the study footprint were the lowest for red imagery (961 m$^2$), while gaps increased in the green (1494 m$^2$) and NIR (1849 m$^2$) clouds (Figure 7e–h). Imagery sets in red edge and NIR also registered numerous tree canopy features in the minimum elevation raster (Figure 7e,g). This results in an increased interpolation distance for ground models where only canopy features have been reconstructed.

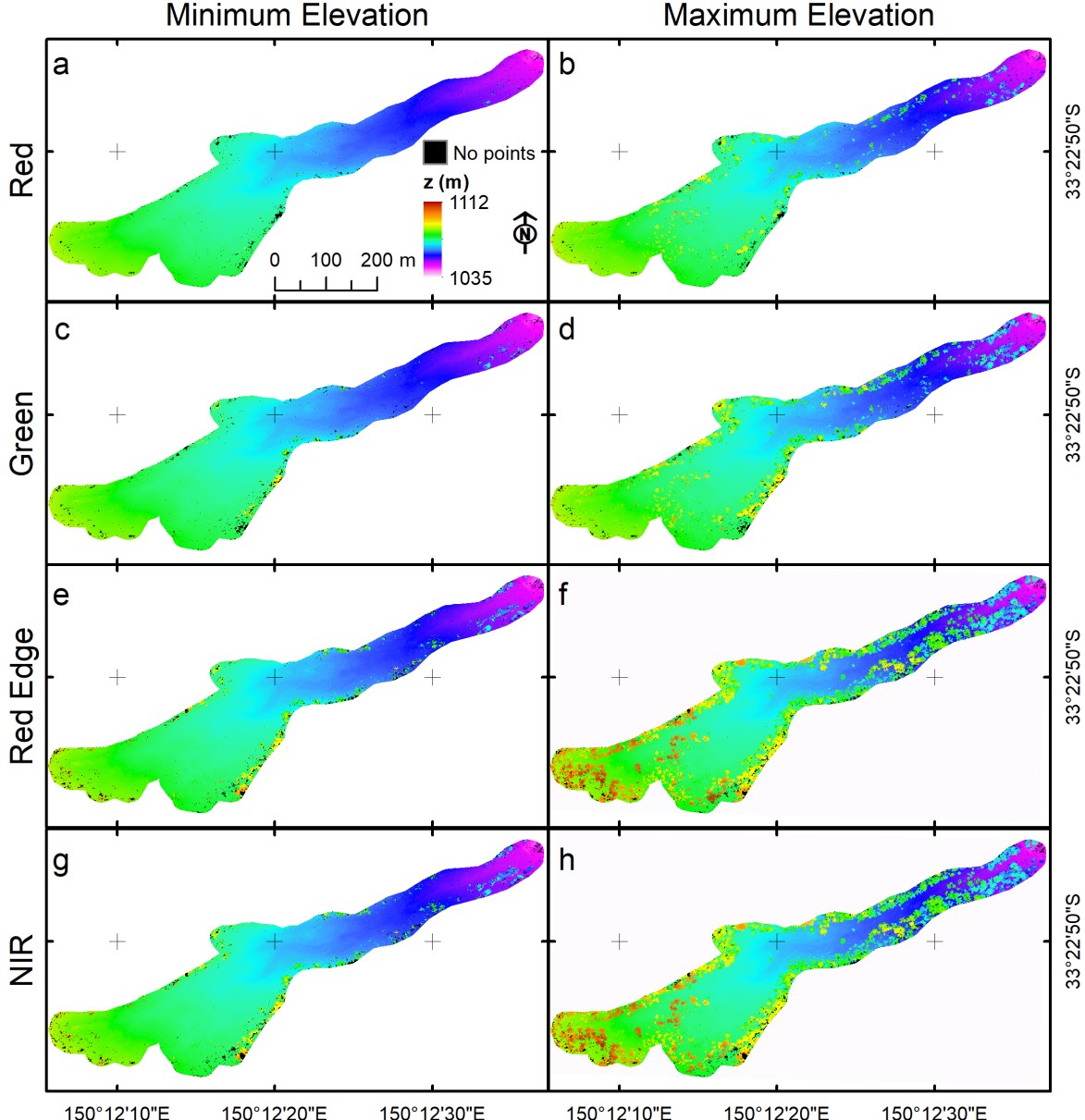

**Figure 7.** Midday summer multispectral imagery processed to dense point cloud stage as individual cameras and sampled at 2 × 2 pixel resolution (0.16 m GSD) for minimum (**a,c,e,g**) and maximum (**b,d,f,h**) dense point elevation.

For single spectral band images collected in summer at T2, total dense points reconstructed followed the order red < green < NIR < red edge, resulting in projects with 38.3, 40.4, 47.6, and 47.7 M points, respectively. Features reconstructed 0–2 m above ground level dominated all dense point clouds, with total points between 34.4 and 36 M in the order NIR < red edge < green < red. This combination contributed to a clear pattern of tree canopy feature reconstruction with wavelength (Figure 8). Less than 6% of red band but more than 27% of NIR band dense point features were greater than 2 m above ground level. In terms of absolute dense point cloud reconstruction greater than 2 m AGL; NIR imagery reconstructed 6.2 times more dense points than red band imagery (12.88 M vs. 2.09 M). The selection of green cameras more than doubled the total dense point features registered above 2 m AGL (4.24 M).

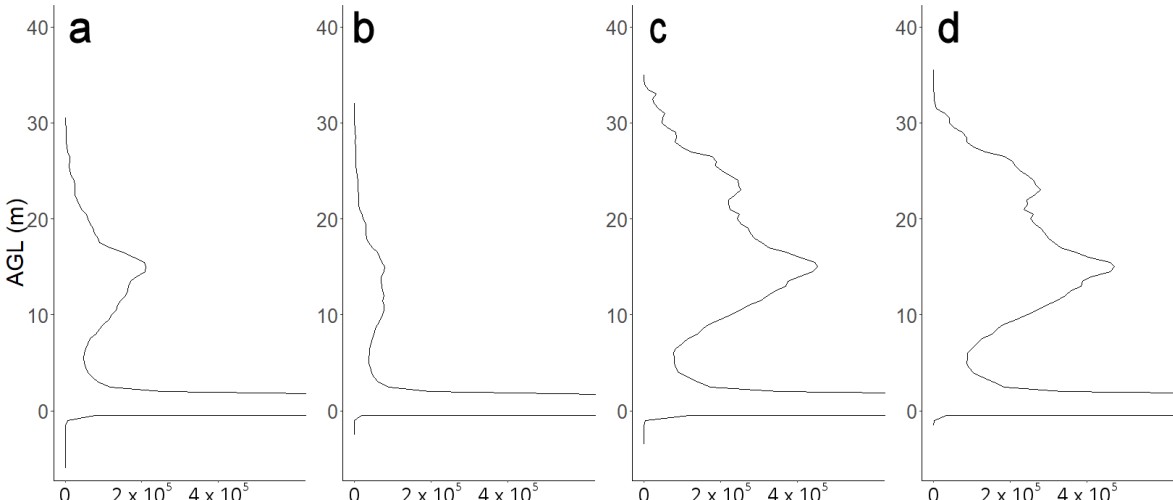

**Figure 8.** Dense point cloud frequency plots at 0.5 m bin widths for (**a**) red, (**b**) green, (**c**) red edge, and (**d**) NIR. Point frequencies less than 2 m have been cropped to improve the interpretation of reconstructed points more than 2 m AGL.

### 3.1.2. Multicamera vs. Single Camera

Processing the T2 green camera alone generated 38.8 M dense points, of which 1.5 M were unique features compared to the same cameras processed as the master camera for a four-band multispectral camera rig. Conversely, the green camera master product reconstructed 37.3 M dense points, of which 548 k were unique to this product. Dense point cloud features more than 2 m AGL were reduced by 1.3 M (4.25 M vs. 3.26 M) by changing processing from green camera alone to green camera as master. The only change in processing was the setting of camera properties to include the other spectral band cameras (Figure 9 main). The green camera alone generated unique dense points from ground level through the tree and tree canopy, with the majority in two bands: 0–2 m and 10–20 m AGL (Figure 9a). The lower total number of unique reconstructed dense point features in the multicamera rig output were spread throughout the vertical range from ground to canopy (Figure 9b). The green camera acting as master for the multispectral camera rig resulted in fewer reconstructed tree canopies; however, several exceptions are clear in the dense point cloud (Figure 9 main). The impact of processing for multispectral orthomosaic preparation (master camera) was compared between the green and NIR cameras by adjusting the camera rig settings. The NIR camera set to master reconstructed almost fourfold (9.6 M) more dense point cloud features than the green camera (Figure 10). The majority of unique points reconstructed by the green camera were ground features (0–2 m AGL) (Figure 10a), while 2.2 M unique features reconstructed with the NIR camera set to master were >10 m AGL (Figure 10b).

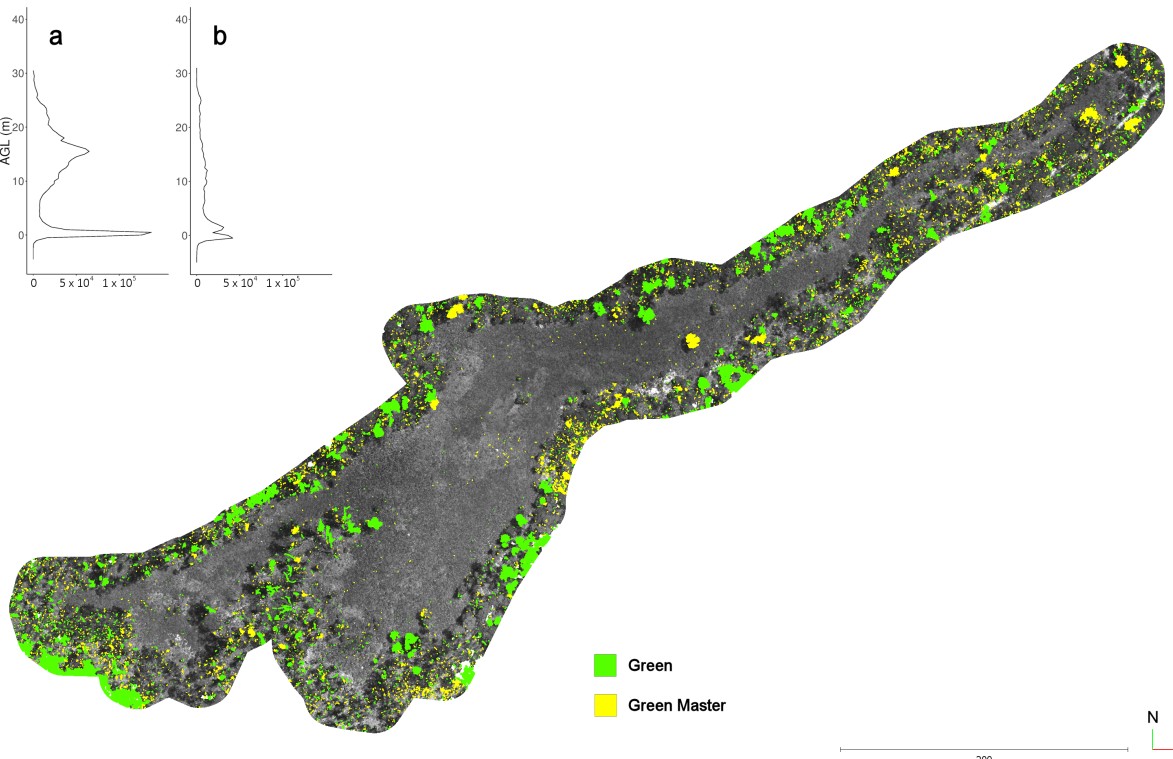

**Figure 9.** (main) Summer T2 dense point cloud differences between the green band alone (green) and the green band as master (yellow), highlighted against dense point features of both products separated by less than 0.5 m. The vertical frequency of unique features in (**a**) the green band only and (**b**) the green band master processed with identical settings.

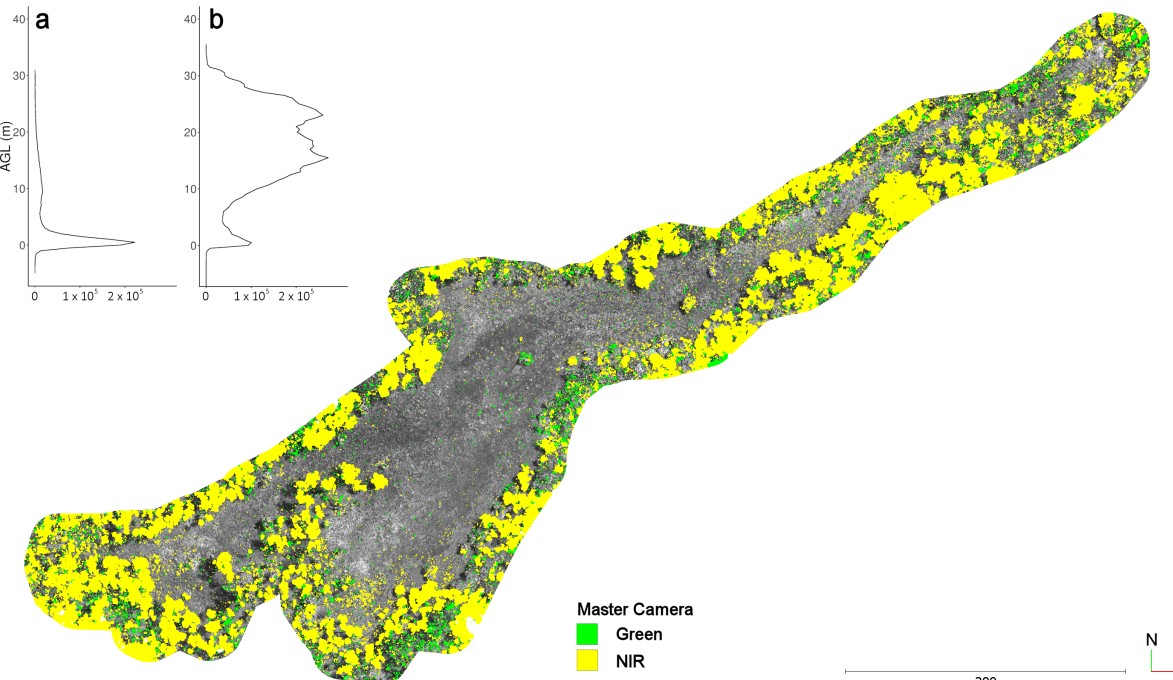

**Figure 10.** (main) Summer T2 dense point cloud features for the green band master (green) and NIR band master (yellow), with dense point features of both products separated by less than 0.5 m. The number of unique dense point elevations above ground level (0.5 m bins) is shown for (**a**) green and (**b**) NIR dense points.

### 3.1.3. Time of Day

Imagery captured at three times of day for the study area were processed independently using the 530–570 nm (green) camera as the master. Morning (T1), midday (T2), and afternoon (T3) resulted in total dense point clouds of 45.5 M, 37.9 M, and 41.2 M respectively. At a separation of 0.5 m, the total of uniquely identified features at T1 (5.44 M), T2 (0.97 M), and T3 (2.42 M) were approximately 12%, 2.5%, and 5.9% of the total points reconstructed in each project, respectively (Figure 11). The vertical distribution of uniquely registered features also differed between times of day, with T1 values distributed at higher elevations (Figure 11a), while at T2 the majority of uniquely reconstructed points represented ground features and at T3 a combination of ground and mid-level features were uniquely reconstructed (Figure 11b,c).

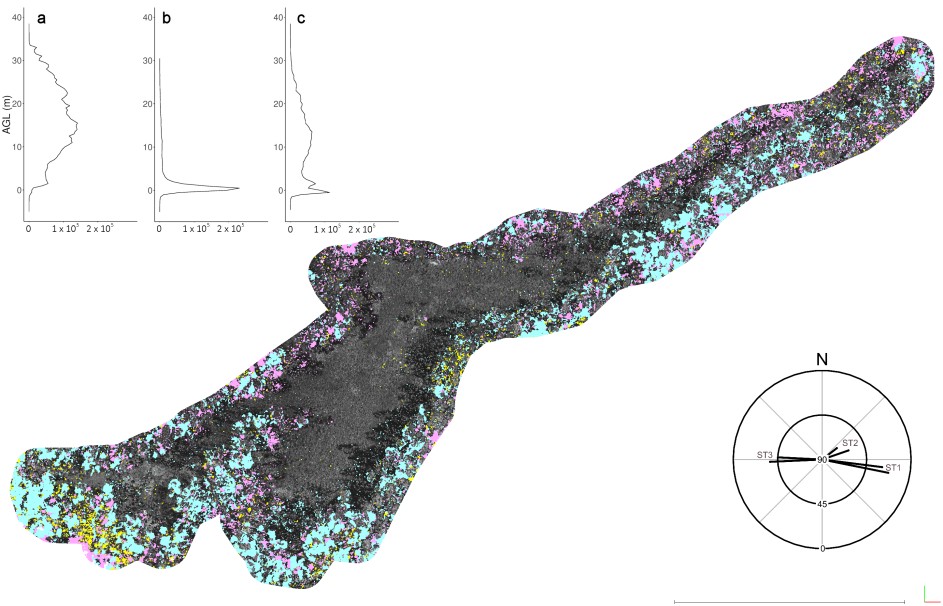

**Figure 11.** (main) Dense points identified in the morning, at midday, and in the afternoon that are more than 0.5 m from the nearest point registered in both other products from imagery sets processed using the green camera as master. Elevation profile (1 m bins) of unique point features for (**a**) morning (T1), (**b**) midday (T2), and (**c**) afternoon (T3). The radial plot shows the start and end solar azimuth and the elevation for the three imagery sets.

Likewise, modifying the camera master to NIR reconstructs more canopy features in the morning and afternoon dense point cloud products. Morning (T1) imagery contributed 5.1% (2.4 M) unique points, while those of the afternoon (T3) contributed 4.1% (almost 2 M) (Figure 12a,b). These features represented both ground and canopy features. More than 4.3 M dense point features were recorded in both the morning and afternoon (< 0.5 m) that were absent in the midday product (9.2%) (Figure 12c). A total of 18.5% of possible dense features were not reconstructed in the midday imagery set. Of these features, more than 78% were located 8–34 m above the ground, while ground features (< 2 m) were less than 8.2%. Of features absent from the midday dense point cloud, approximately 50% were reconstructed in both the morning and afternoon, while the remaining 50% were unique to either the morning or the afternoon (Figure 12 main).

Winter season imagery capture was constrained to two times of day at different solar azimuths but similar solar elevations (Figure 13). Reconstructed dense point cloud features followed similar patterns in total dense point number across the site (Figure 13a,b). When uniquely reconstructed points are analysed, early afternoon showed a small shift towards taller canopies (> 20 m) with fewer ground

features (0–2 m) (Figure 13c,d). A slightly higher representation of taller tree canopies in the afternoon (T2) processed point clouds is confirmed in Figure 13 (main).

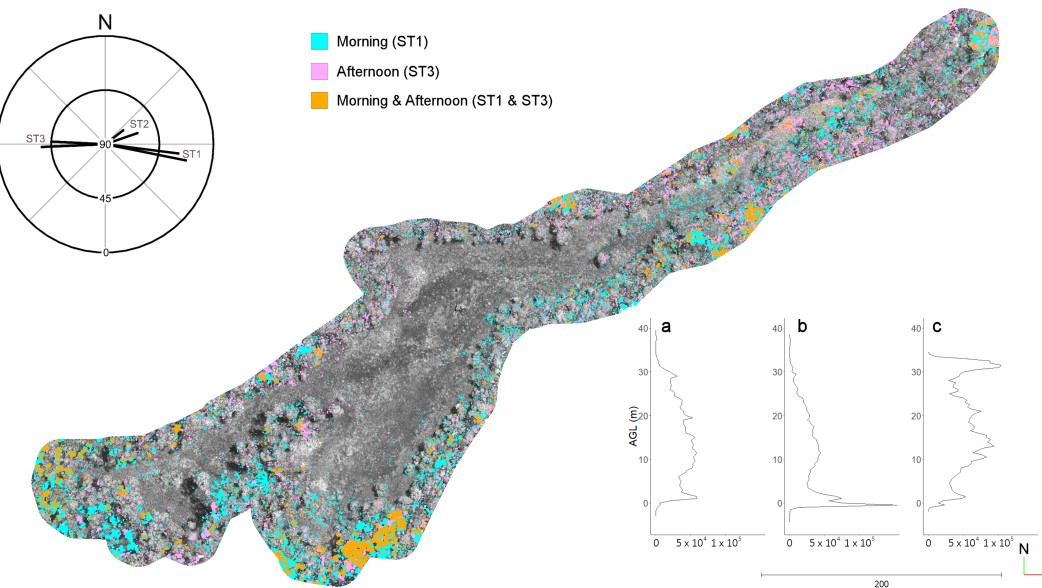

**Figure 12.** (main) Dense point cloud differences for NIR band master imagery products collected in summer. The solar elevation and azimuth at time of first and last images collected for T1, T2, and T3 are shown for reference. Blue points are features uniquely reconstructed in T1 imagery, with the (**a**) corresponding vertical frequency plot. Pink points are features unique to T3 imagery (**b**), and orange points are features recorded in both T1 and T3 products (**c**). Background dense points were generated at T2.

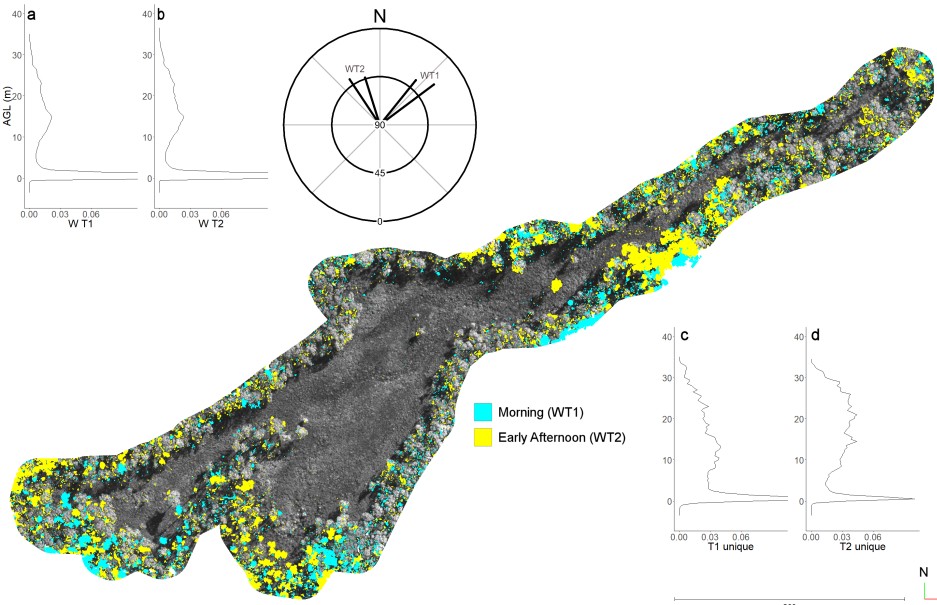

**Figure 13.** (main) Dense point cloud differences for NIR band master imagery products collected during winter. Solar elevation and azimuth at the time of first and last images collected for T1 and T2 are shown for reference. Blue points are features uniquely reconstructed in T1 imagery, with (**a**) the corresponding vertical frequency plot. Yellow points are features unique to T2 imagery, with (**b**) the relative frequency plot for the vertical distribution of points. Relative frequency of dense uniquely reconstructed dense points at (**c**) T1 and (**d**) T2 for points are separated by more than 0.5 m from the independently created point cloud.

### 3.2. Reproducibility

Solar geometry varies throughout the year, so that it is possible to align solar azimuth or elevation but not both in different seasons. Reproducibility was assessed by comparing the outputs of single time points using an NIR camera as master for similar solar azimuth or elevation range. The imagery sets in this project included winter and summer conditions where either solar azimuth or elevation were similar. Dense point cloud reproduction with varied solar elevation was examined using midday summer (ST2) and winter morning (WT1) where azimuth overlapped (Table 1). To directly visualise tree canopy changes, underlying terrain was removed from dense point clouds using the 2 m minimum elevation decimated red camera product from ST2 (Figure 7a). Increased solar elevation resulted in the registration of 1.2 M unique features < 2 m AGL (31%) (Figure 14a), while low elevation (WT1) contributed 0.5 M (11%) dense points (Figure 14b). Dense points representing tree canopies (> 10 m AGL) were more numerous (0.3 M) at low solar elevation (WT1) (Figure 14b). Numerous tree canopies are visible that have been reconstructed in only one product (Figure 14 main). Tree canopies also show reconstruction of additional shaded canopy elements with higher sun elevation in ST2. The profile of relative frequency in WT1 (Figure 14b) between 0–20 m AGL indicates reconstruction of tree trunks and branches as well as lower canopies (2–10 m AGL).

The reproducibility of tree canopy between seasons with equivalent solar elevation ranges were assessed between afternoon imagery sets for winter and summer (Figure 15). Total dense point cloud elevation profiles for summer afternoon (ST3; 44.1 M ) and winter afternoon (WT2; 45.7 M) show very similar point cloud profiles (Figure 15a,b). Both dense point clouds contained approximately 8% uniquely reconstructed features: ST3 (3.8 M) and WT2 (3.7 M), respectively. For both ST3 and WT2, the majority of these dense points were > 2 m AGL (67% and 79%, respectively). Uniquely reconstructed dense points more than 10 m AGL at ST3 (1.8 M) and WT2 (2.2 M) were distributed (> 10 m AGL), and ground (0–2 m AGL) features that were not consistently reconstructed between seasons when solar elevation ranges were similar.

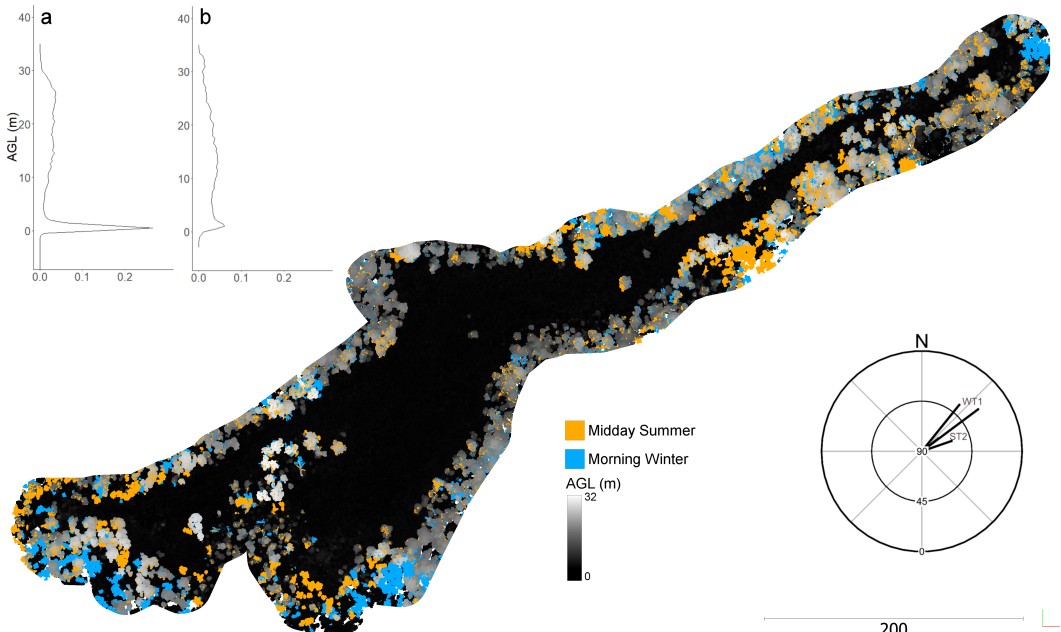

**Figure 14.** (main) Dense point cloud reconstruction of features separated by < 0.5 m in both products combined and displayed as a height ramp. Dense points > 10 m AGL that are uniquely reconstructed are shown projected so that the highest AGL points are visible. Elevations for all products are calculated as z distance component from a red point cloud 2 m z minimum subsample to remove underlying terrain. The relative frequency of uniquely reconstructed point features for (**a**) ST2 and (**b**) WT1.

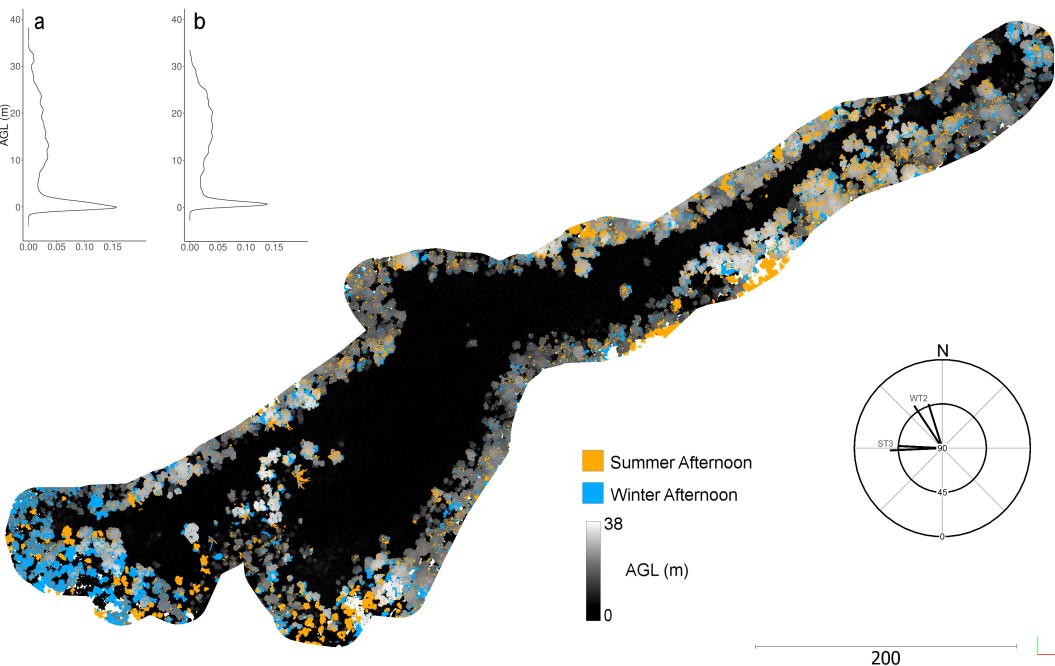

**Figure 15.** (main) Dense point cloud reconstruction with ST3 (red to yellow) and WT2 (blue) compared to features separated by < 0.5 m in both products combined and displayed as a height ramp (black–white). Elevations for all products are calculated as z distance component from a red cloud 2 m minimum subsample to remove underlying terrain. The relative frequency of uniquely reconstructed point features for (**a**) ST3 and (**b**) WT2.

Winter T1 and T2 dense point clouds were merged and compared with the merged product of summer T1 and T2. Total dense cloud counts were 89.5 M and 93.8 M, respectively. Summer and winter seasonally unique features represented less than 4% of total dense point clouds, with 3.74 M and 3.37 M, respectively (Figure 16). The larger solar elevation range during summer resulted in more uniquely reconstructed ground features than winter (0.93 M vs. 0.48 M) (Figure 16a). Summer also reconstructed a greater AGL range than winter (38.4 m vs. 33.1 m AGL) (Figure 16b). Combining WT1 and WT2 resulted in a total of 29 M dense points reconstructed more than 2 m AGL, while summer (ST1 and ST2) returned 30.4 M. In both cases, this represents 32.5% of the total dense points reconstructed.

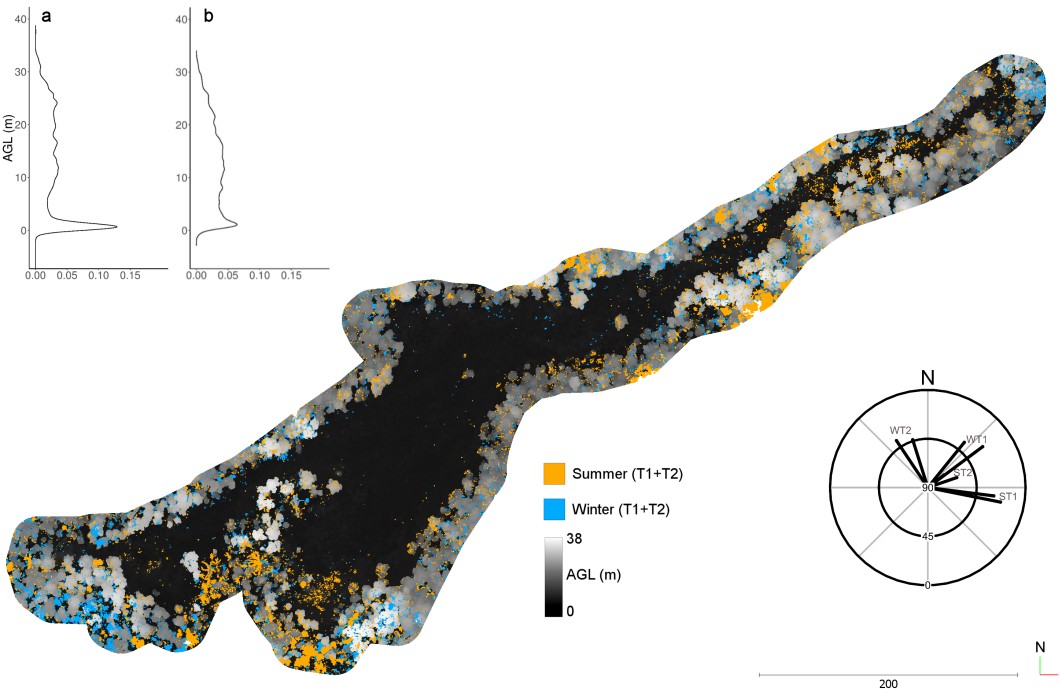

**Figure 16.** (main) Dense point clouds of reconstructed points within 0.5 m distance to alternate season point cloud (black to white). Unique points in summer (orange) and winter (blue). Relative frequency at different heights AGL for (**a**) summer and (**b**) winter.

## 4. Discussion

The properties of the densified point cloud impact all subsequent outputs, including digital surface models and orthomosaics. Within a single flight near solar zenith, imagery collected simultaneously reproduced a fourfold difference in projected tree canopy. Tree canopies are complex photogrammetric targets and, in this study, introduce steep elevation changes at canopy boundaries that may create artefacts in subsequent raster products [30]. It is important to establish sources of variability in dense feature registration if reproducible canopy models are to be created, particularly in subtropical and tropical regions where water or nutrients rather than light limit tree canopy [10]. Assmann and coworkers [18] provide a review of factors associated with radiometric correction and the collection of comparable multispectral imagery for high latitudes. They note that radiometric correction is conducted in the software packages such as that used in this paper but that the process remains a black box [18]. The image feature generation, matching, and densification algorithms are also proprietary, resulting in uncertainty in the outputs.

### 4.1. Camera Target Geometry

Small UAS operations are legally restricted to less than 120 m above ground level in Australia and many other countries [55]. Camera field of view and flight altitude determine ground sample distance, with increasing altitude corresponding to lower spatial resolution [56]. Flying at lower altitudes results in smaller image footprints and decreased forward and side lap intervals [56]. Regardless of final flight altitude, tree canopies will represent at least 30% of the total distance between ground and sensor. Establishing a monitoring program requires a range of compromises to allow data collection, processing, and reporting in a timely manner. In addition, data must be reproducible and transferable as monitoring programs are often carried out by different people over time [6]. The Parrot Sequoia used in this project is an accessible and self-contained sensor that can be carried on a wide range of platforms due to its small size and weight. A flight altitude of 80 m in this study resulted in a sub-decimetre pixel ground resolution while allowing site coverage in three 10 min flights. Tree canopy height varied across the forest areas in the study site, but some individuals reached approximately

40 m height. This results in pixel resolution decreasing by approximately 50% to 4 cm, while overlap proportion decreases to 70% side lap and 75% forward lap (48 m × 37 m). Tree canopy features that poorly adhere to photogrammetric requirements are therefore present in fewer images at wider perspective ranges. Employing red edge or NIR spectral imagery improves the reconstruction of upper canopy features under these conditions.

### 4.2. Spectral Band and Camera Master

The selection of spectral band determines the nature of features available for matching. For multispectral cameras, each spectral band will generate different patterns of registerable features with different locations within the scene. This is particularly relevant for high vegetation coverage targets where the sensor is built to maximise the range of reflectance in green vegetation (red, green, red edge, NIR). The digital surface model must be common for all spectral bands to allow sensible orthorectification and blending of images so a single master band is required. Even with interacting spatial resolution and overlap, NIR and red edge imagery reconstructed fourfold more tree canopy projected cover. Tree canopy feature registration appears to increase with spectral reflectance of leaves from red > green > red edge > NIR. Increased tree canopy reconstruction in the NIR band is achieved at the expense of ground layer modelling. Red spectral band processing of air survey block near nadir imagery close to solar zenith largely excluded tree and canopy features. At this site, a discontinuous and porous Eucalyptus canopy cover allowed near complete modelling of ground by selecting elevation minima at 2 m spacing. The reconstruction of canopy height models in undulating terrain is likely to be possible in this instance.

### 4.3. Time of Day

The summer midday imagery products had the lowest reconstruction of tree canopy features within any spectral band; however, visible bands, red and green, performed poorly compared to red edge and NIR. Low sun elevations (∼25°) improved the reconstruction of tree canopy features in dense point clouds regardless of spectral band. Low sun elevations resulted in increased point reconstruction throughout the tree profile and maximum height recorded for tree canopy. Improved registration of canopy features at this site may be the result of pendulous leaf orientation of most trees. Low solar elevations will be closer to perpendicular with leaf surfaces while high solar elevation incident is closer to parallel. In summer, solar azimuth covered more than 180° but did not consistently increase canopy dense point cloud feature reconstruction.

Combining Dense Point Cloud Products

In the case of natural Eucalyptus tree canopy structures, capturing imagery at two different times of day improved the reproducibility of canopy models. This was achieved by increasing the total dense point number from 40–45 M to 85–90 M. It was possible to achieve approximately equivalent total dense point features for tree canopy when two dense point cloud products were combined. However, there remained approximately 4% of dense canopy features that were inconsistently reconstructed; these included dead tree stags and young saplings. The majority of community boundary canopy is reproducible, and many of the features separated by more than 0.5 m are canopy edges or intra-canopy gaps. In this study, the imagery capture window was restricted by access and safety requirements. This is a common constraint in remote locations and not a reason to exclude this site from study.

### 4.4. Digital Terrain Models

The choice of spectral band master or processing of individual spectral bands results in registration and reconstruction of different features in forest environments. In this study, red band imagery processed as a single camera and collected at the highest solar elevation achieved near complete coverage of ground and near-ground features. Processing red band imagery limits the reconstruction of tree features and also results in more complete ground-level feature registration. Occlusion or

vertical uncertainty are typically introduced as a result of vegetation cover in geomorphological studies. Processing requirements are further reduced with this approach. Future studies should assess the impact of camera station density and ground resolution on subsequent ground-model reconstruction in red and blue spectral bands where vegetation reflectance and subsequent features are suppressed.

### 4.5. Vegetation Models

### 4.5.1. Shrub and Tussock Grasses

The study site is defined by statutory mapping products and consisted of approximately equal areas of forest canopy and shrub vegetation. Shrubby vegetation was consistently modelled in all spectral bands at the spatial tolerance and spatial resolutions used in this study. Shrubby vegetation presents a complex surface when assessing illumination, but movement in wind is limited and the variation in distance from sensor to target is limited [18]. As this vegetation is typically less than 2 m tall, it also falls within the coarsest spatial resolution of captured images (approximately 8 cm). Crops are also likely to present a similar uniform elevation where default processing is appropriate due to viewing geometry [9]. Shrub communities present a robust target for UAS remote sensing for these reasons; however, occlusion by Eucalyptus canopy remains a challenge for community boundaries.

### 4.5.2. Trees

Registration of smaller Eucalyptus and other sparse canopied trees has previously been unsuccessful [8,47], and no publications address the reproducibility of tree canopy models with time of day or single spectral band selection. Goldbergs found that canopy reconstruction was improved when an RGB camera was modified to include NIR [42]. Time of day, processing settings, and spectral properties of the tree canopy may all contribute to the failure to model these features. Choosing NIR and low sun elevation resulted in the reconstruction of small tree canopies in this project. The development of woody vegetation may be an important target for assessment [7,8], but unless imagery is captured in a manner that maximises canopy features, it is unlikely to be reconstructed. Low sun elevations improve tree canopy registration if green or red band imagery is used as the master camera. Canopy structure is more complete when master cameras are NIR or red edge spectral bands, but a range of sun illumination geometries are still required to improve canopy reconstruction.

Differences in dense point cloud feature reconstruction propagate to differences in digital surface models and subsequent orthomosaic products. Increasing the number of total features reconstructed generally improves the number of tree features reconstructed. However, for an identical camera station set, using NIR or red edge spectral bands rather than visible bands can improve registration of tree canopy features at the cost of ground feature registration. Default settings for the software used in this study are a green band master, which is likely a compromise suited to targets containing a mixture of built and natural features. Applying an NIR or red edge spectral band as master or sole camera when studying canopy structure or change detection in orthomosaic products of complex canopies appears to improve reconstruction of target features. Low solar elevations are typically avoided in remote sensing, with solar noon targeted to reduce shadow and improve radiometry (e.g., [57,58]). It has been previously demonstrated at lower spatial resolutions that reconstruction is possible at low solar elevations for large-format digital imagery [59]. We found that tree canopy models were improved by the collection of imagery at low solar elevations in all spectral bands. The fundamental process of detecting and matching features may be enhanced in visually porous canopies such as natural Eucalyptus when low solar elevation improves reflectance from vertically oriented leaves. Low solar elevations also resulted in the reconstruction of more 3D features and a higher proportion of tree and canopy features. This may be due to the improved contrast between canopy and ground resulting from shadow.

*4.6. Software Effects on Reconstruction*

In this study, software filtering associated with multicamera rig processing resulted in consistently fewer dense points compared to single-camera processing. The reason for additional filtering of reconstructed dense points in master band camera settings is not clear, although the effect is consistent regardless of the spectral band employed. Underlying variability in products, noted in particular with the default green spectral band, represents a base level of variability in products that will negatively impact change detection through time.

*4.7. Reproducible Products for Change Detection*

Monitoring and change detection rely on repeatable and reproducible products [6]. Inconsistent reconstruction of features in dense point clouds, whether caused by software, environment, sensor, or sensor placement, will propagate to raster products and reduce the ability to detect change. Combining imagery collected at multiple solar elevations/azimuths improves the completeness of canopy reconstruction while reducing proportional error rates between image epochs. The potential of UAVs to collect hypertemporal imagery sets is ideally suited to developing more complete vegetation models where approaches mitigate the following caveats.

1. Monitoring and change detection rely on repeatable products. This is not possible with default settings or single-time-of-day products for Eucalyptus tree canopies.
2. Variable reconstruction of tree canopy features affects the interpretation of results and all subsequent photogrammetric products. This is likely to reduce change detection sensitivity.
3. Caution should be exercised when interpreting single image set case studies for which the implicit assumptions are either that (a) provided the environmental conditions are replicated and/or (b) that a specified protocol is followed, the resulting products will be reproducible.

Although tree canopy features represent less than 40% of total reconstructed points at this site, the impact on subsequent raster products is substantial due to the sharp elevation changes introduced at the canopy boundary and deeply convoluted surfaces in canopy gaps. Reproducibility of tree canopy structure is essential for monitoring programs and may be improved by combining imagery collected at multiple solar elevations and azimuths. In this study, NIR imagery increased the projected footprint of reconstructed canopy fourfold over red and 2.3-fold over the default green spectral band. Canopy and subsequent modelled tree characteristics that apply canopy height models should consider the essential target properties required for subsequent analyses and employ the hypertemporal capacity of UASs to capture imagery containing features associated with target metrics.

## 5. Conclusions

The UAS image photogrammetry workflow is complex [14] and fundamentally dependent on image feature extraction and registration. We demonstrate for Eucalyptus canopy that both environmental and processing decisions inherently impact the final densified 3D models. Proprietary software limits the ability to understand some processing outcomes, but the following were consistently found in this study.

- Tree canopy features were poorly reconstructed using green and red spectral imagery collected at high solar elevations, but completeness of ground feature reconstruction was improved. It is possible to generate digital elevation models with minimal filtering and high coverage using this imaging approach.
- Conversely, tree canopy models are improved when NIR and red edge spectral bands are used as the camera masters for processing multispectral imagery.
- Combining imagery collected at low and high solar elevations improved the completeness and reproducibility of vegetation models, with less than 4% variation in reconstructed features.

- Shrub communities represent robust and repeatable photogrammetric features at the spatial and spectral resolutions applied in this study. Small Eucalyptus tree saplings were variably reconstructed and dependent on solar elevation and master camera choice.

While small UASs provide potential for hypertemporal resolution imagery capture, we find that this rapid recapture of imagery may be necessary to allow the effective reproduction of Eucalyptus canopy structure and detection of saplings.

**Author Contributions:** Conceptualization, R.M. and A.F.; methodology, R.M. and A.F.; validation, R.M.; formal analysis, A.F.; investigation, R.M.; data curation, A.F.; writing—original draft preparation, A.F.; writing—review and editing, R.M.; visualization, A.F.; funding acquisition, R.M. and A.F. All authors have read and agreed to the published version of the manuscript.

**Funding:** Work presented in this paper was funded by the Australian Coal Association Program Project C25056.

**Conflicts of Interest:** The authors declare no conflict of interest. The funders had no role in the design of the study; in the collection, analyses, or interpretation of data; in the writing of the manuscript, or in the decision to publish the results.

## Abbreviations

The following abbreviations are used in this manuscript:

| | |
|---|---|
| FWHM | Spectral resolution expressed as full width (of wavelength range) at half maximum (of spectral response intensity [DN/%]) |
| GNSS | Global Navigation Satellite System |
| UAV | Unmanned aerial vehicle |
| UAS | Unmanned aerial system |
| FOV | Field of view in degrees |
| RGB | Red green blue colour visible colour image |
| CSD | Camera station density |
| AGL | Above ground level or launch point for UAV used |

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
