# Peer review of "Hypertemporal Imaging Capability of UAS Improves Photogrammetric Tree Canopy Models"

_remotesensing, doi:10.3390/rs12081238_

Round 1

Reviewer 1 Report

The paper presents the results from an experiment at a single site: to acquire multiple UAS data using different sensors, at different times of day and year, and to process the data using different parameters, with the aim of testing the reproducibility of derived 3D point clouds. This is a valuable experiment, which determined some important conclusions that would be of interest to many of Remote Sensing’s readers. However, several important aspects of the paper were poorly conducted, making what should have been an interesting read into a difficult slog. Each of these major issues are described below, followed by a list of minor problems. Fixing the problems will require a major rewrite, though the revised document will be significant, given the nature of the conclusions.

Major problems

  1. The introduction and description of the method of producing the point clouds is inadequate. You mention UAV image processing “pipelines”, but you don’t describe them, or have a flow chart that shows what they commonly involve. You mention “final products” and “resulting products” but you don’t describe what they are or how they are produced. The main description of the image processing method in section 2.4 is extremely inadequate. This is the main method being tested, and is very important to the research, but does not state how Pix4D works, or what the parameters listed in Table 3 are. Even if you don’t know everything about the method because it is proprietary software, you still need to describe a summary.

  1. The choice of analyses conducted on the point cloud (section 2.5) were poorly explained and have no clear purpose. Each of the steps needs a careful description and justification. Why did you calculate Euclidean distance, classify distinct features, filter noise, classify inconsistently reconstructed features, perform rasterization (to what products?), and 2 m spatial decimation?

  1. The review of the literature in the introduction is strangely worded, and many sentences should be reconsidered. For example, the following sentence is difficult to follow:

“Canopy and some ground features were reconstructed in close range photogrammetry (<15m) using an RGB camera, but when compared with UAV LIDAR, upper canopy branches and sub-canopy ground features were occluded [24].”

This should be rewritten to:

“When canopy and some ground features were reconstructed in close range photogrammetry (<15m) using an RGB camera, upper canopy branches and sub-canopy ground features were occluded when compared with UAV LIDAR [24].”

The following sentences were also problematic:

“In natural savannah ecosystems larger trees were modelled using an RGB modified to remove the IR filter. Larger tree canopies achieved 70% identification compared to LIDAR but smaller canopies were poorly modelled [23].”

And should be changed to:

“Although larger tree canopies in natural savannah ecosystems have been successfully modelled using data captured by an RGB sensor modified to remove the IR filter smaller canopies were poorly modelled [23].”

Also, the following sentence is unclear, and I don’t know what the point is you are attempting to get across.

“In this study, dissimilarities between dense point clouds, particularly those of tree canopy features are considered they may disproportionally affect final products.”

As this is a repeated issue, I have included it as a major problem.

  1. There are also several major problems with the figures that need to be fixed as they are such an important component of the work. Figure 3 is very hard to interpret without colours, as the size differences between points are too small. Figure 4 needs column and row labels, as the reader has to constantly refer back to the caption, which makes interpretation very difficult. The maps need colour legends and clearer captions (I don’t know whether the green and yellow pixels in figure 6 overlap or not?). You don’t refer to figure 9, 10 or 12 in the text, so I don’t know why they are included.

Minor problems

  • Many acronyms are presented without explanation. Even if they are commonly used, they should be defined properly. For example, UAS, UAV, GNSS, 2D, 2.5D, 3D, RGB, NIR, DSM, EEC, GPS, IMU, etc.
  • The mention of the “NIR band camera master” in the abstract (line 9) needs some explanation as this is not a commonly understood phrase.
  • The sentence on line 103 starting “Natural broadleaved forests...” does not fit with the rest of paragraph and could be deleted. The “range of biophysical properties” mentioned has nothing to do with the acquisition and processing parameters being tested.
  • The second aim is strangely worded and needs to be changed. You are not extracting multi-camera rigs. Also, what are the resulting products you mention?
  • The sentence on line 126 starting “Collecting imagery…” is repeated from aim 1, has nothing to do with the choice of spectral band, and should be deleted.
  • The site selection and the role of the fire is not well explained. If the main need for detecting reliable changes in canopy heights from multi-temporal UAS data is to detect tree encroachment into shrub/sedge/grass environments, then this should be discussed more, and the focus of the aims, discussion and conclusion should change.
  • The sentence on line 150 starting “Wind observations…” is unnecessary and should be deleted. However, you need to include in the caption to Table 1 that the wind measurements were recorded in Lithgow.
  • I don’t know what you mean by the sentence “Seasonal difference in camera station location is equivalent to 50% survey line offset…” or why this sort of calculation is relevant.
  • Did you mean to add some references to the sentence on line 334 where you wrote (REFS)?
  • When you write “this attribute” on line 426 wat attribute are you referring to?

Reviewer 2 Report

peer-review: remotesensing-732659

Dear Authors,
congratulation to you very interesting manuscript.
Your research has successfully investigated several of the factors that affect reconstruction of tree canopy structure in natural forests to identify methods for developing reproducible tree canopy models.
Your manuscript is scientifically well designed, soundly written and well presented. Your paper is up-to date and fits well into the scope of the journal considered.
However, as many manuscripts require some correction and enhancements also your paper can still be improved.
You will find my comments, suggestions and recommendation for a revision displayed with the Adobe commenting tools in the review version of your manuscript attached.

Round 2

Reviewer 1 Report

All my comments have been addressed, and the paper has improved considerably. I still found that was difficult to read, with overly long sentences, inconsistency of tense, and figures that were difficult to interpret. However, I cannot devote any more time to suggesting improvements in the current circumstances, so perhaps it is OK to publish. The results should be of interest to the readers of Remote Sensing.